# Epidemiology of Self-Reported Diabetes Mellitus in the State of Maranhão, Northeastern Brazil: Results of the National Health Survey, 2013

**DOI:** 10.3390/ijerph16010047

**Published:** 2018-12-25

**Authors:** Rafael Alves Guimarães, Otaliba Libânio de Morais Neto, Marta Rovery de Souza, Juan José Cortez-Escalante, Thays Angélica de Pinho Santos, Claci Fátima Weirich Rosso, Márcio Mangueira Pacheco, Jamesson Ferreira Leite Júnior, Guthardo Sobrinho França, Lilia de Jesus Fonseca, Ludmila Grego Maia

**Affiliations:** 1Instituto de Patologia Tropical e Saúde Pública, Universidade Federal de Goiás, Goiânia 74605-450, Goiás, Brazil; otaliba.libanio@gmail.com (O.L.d.M.N.); martary@gmail.com (M.R.d.S.); 2Organização Pan-Americana da Saúde, Oficina Nacional no Brasil, Brasília 70800-400, Distrito Federal, Brazil; cortezj@paho.org; 3Faculdade de Enfermagem, Universidade Federal de Goiás, Goiânia 74605-080, Goiás, Brazil; thays_angel_8@hotmail.com (T.A.d.P.S.); claci.fen@gmail.com (C.F.W.R.); 4Secretaria do Estado da Saúde do Maranhão, São Luís 65076-820, Maranhão, Brazil; mmpsousa2010@hotmail.com (M.M.P.); jjpsi@hotmail.com (J.F.L.J.); guthardosf@gmail.com (G.S.F.); licafonse@bol.com.br (L.d.J.F.); 5Curso de Enfermagem, Universidade Federal de Goiás, Jataí 75804-020, Goiás, Brazil; lgregomaia@yahoo.com.br

**Keywords:** epidemiology, diabetes mellitus, diabetes, risk factors, prevalence

## Abstract

*Objective*: To estimate the prevalence and risk factors for self-reported diabetes mellitus (DM) in adults from the State of Maranhão, Northeastern Brazil. *Methods*: A cross-sectional study was carried out with 1774 individuals aged ≥18 years participating in the National Health Survey of 2013 in Maranhão. The adults were selected by probabilistic sampling and interviewed face-to-face by in-home visits. The Poisson regression model was used to verify the factors associated with DM. *Results*: The prevalence of DM was 5.39% (95% confidence interval [95% CI]: 3.73–7.73). After adjustment of the regression model for age, gender, smoking, education, hypertension, and hypercholesterolemia, DM was statistically associated with age ≥60 years, female sex, low educational level, and self-report hypertension. *Conclusion*: The present study found the prevalence of self-reported DM similar to that estimated in the general population of Brazil. Public policies for prevention and control should intensify control, especially in the subgroups most vulnerable to DM.

## 1. Introduction

Diabetes is a serious public health problem throughout the world, representing one of the most important epidemic diseases of the century [1,2]. In 2014, the World Health Organization (WHO) estimated the global diabetes prevalence to be 8.5% of the adult population, a frequency almost two times higher than in 1980 (4.7%) [2]. This variation stems from the increase in population aging and the prevalence of classical risk factors such as inadequate eating habits, physical inactivity, and obesity [2,3]. It is estimated that this increase was responsible for approximately 1.5 million deaths worldwide in 2012 [2]. Based on disability-adjusted life years (DALYS), 89 million-years of lost life are related to diabetes mellitus (DM) [4]. In 2014, approximately half of all adults with diabetes resided in five countries: China, India, United States, Indonesia, and Brazil [2,5].

In the Region of the Americas, the prevalence of adult DM is estimated to be 8.3% [2], and a large number of them live in Brazil. In 2013, a National Health Survey (NHS) of Brazilian residents estimated the self-reported prevalence of adult DM to be 6.2% [6]. In 2015, the Global Burden of Disease (GBD) identified Brazil as having a high burden of diabetes disease. Specifically, the GBD determined the diabetes case rate for men was 2448.85/100,000 and for women was 1863.90/100,000 [7]. In 2016, a surveillance study of risk factors and prevention measures for chronic disease was conducted by telephone survey (Vigitel in portuguese) in the country’s capital cities, and it identified a self-reported prevalence of DM 9.8% [8]. The Vigitel telephone survey data showed that self-reported DM increased from 4.7% to 7.4% between 2006 and 2012 [9], respectively.

Population-based studies show an association between DM and multiple modifiable and non-modifiable risk factors [6]. The main sociodemographic factors are: increase in age, low education [6,10], and being married. [10]. The main determinants of risk are comorbidities with arterial hypertension (AH), hypercholesterolemia, [10] obesity [10,11], and family history of DM [12]. In addition, low levels of physical activity, inadequate diet, and alcohol and tobacco use have been positively associated with increased risk of diabetes mellitus [10,11,13]. 

In 2013, the WHO set targets for controlling and preventing chronic, non-communicable diseases (NCDs) by 2025. This includes a 25% relative reduction of premature mortality caused by NCDs, including diabetes and will seek to reduce DM risk factors such as physical inactivity, alcohol, and tobacco use [14]. In this context, periodic surveys that monitor diabetes epidemiology and risk factors are fundamental for awareness of trends and patterns in the population’s health that will inform the planning and evaluation of public health policies and prevention programs [15,16]. The objective of this study was to estimate the prevalence of DM and self-reported risk factors of adults in the State of Maranhão, Northeastern Brazil.

## 2. Materials and Methods

### 2.1. Design, Sampling, and Data Source

This is a cross-sectional study using data from the NHS conducted in 2013 [17,18,19]. The NHS was a household survey conducted of the adult population, ≥18 years old, carried out by the Ministry of Health, *Fundação Oswaldo Cruz* (Fiocruz), and the Brazilian Institute of Geography and Statistics. The objective of this study was to produce national data on the condition of the nation’s health, lifestyle habits, and health care service use and access, and preventive actions [17,18,19].

The survey utilized cluster sampling which involved three stages of recruitment. The primary sampling unit (PSU) was composed of census sectors selected by simple, random sampling [17,18,19]. The secondary sampling unit (SSU) was composed of permanent, private households defined as those used for housing one or more persons [17,18,19]. Household selection was performed by simple random sampling (SRS). Finally, a resident aged ≥18 years within each selected household was selected at random to participate in the study, which corresponded to the tertiary sampling unit (TSU). The draw was carried out based on a list of eligible individuals [17,18,19]. The present study analyzed data for the State of Maranhão, located in the Northeast Region of Brazil. 

### 2.2. Investigated Variables

#### 2.2.1. Dependent Variable

Self-reported DM was obtained according to the following question: “Has any doctor ever diagnosed you as having diabetes?” Thus, the prevalence indicator was obtained by the number of participants who positively answered the question from the total number of adults interviewed in Maranhão [20]. Women who reported diabetes that was associated with gestation were excluded from the definition [11]. 

#### 2.2.2. Independent Variables

The following independent variables were analyzed: 

(i) Sociodemographic: Age group, stratified as 18–29 years, 30–39 years, 40–59 years, and ≥60 years; gender, such as proxy of sex (male or female); marital status (living with or without a partner); ethnicity/self-declared skin color (white, brown, black or others [yellow or indigenous]), and education level (no education/incomplete primary education, complete primary education, complete or incomplete secondary education, or incomplete higher education/complete higher education); area of residence (capital city, metropolitan region, or other areas); and type of area (urban or rural). 

(ii) Substance use: Tobacco use (never, former smoker, or current smoker), alcohol consumption in the last 30 days (no or yes) and alcohol abuse in the last 30 days (no or yes), defined as consuming five or more doses of alcoholic beverages on a single occasion for men, and four or more doses on a single occasion for women. 

(iii) Eating habits: Recommended consumption of fruits and/or vegetables (no or yes), defined as a daily intake of at least 400 grams of fruits and vegetables, consistent with the WHO, which is nearly equivalent to consumption of five servings or more per day [21]. The questions used to calculate this variable were: (a) In general, how many times a day do you eat lettuce and tomato salad, or salad of any other raw vegetables or vegetables? (responses: 0 times a day, 1 time a day [at lunch or dinner], 2 times a day [at lunch and dinner], or 3 times or more a day); (b) In general, how many times a day do you eat raw vegetables or cooked vegetables such as cabbage, carrots, chuchu, eggplant, zucchini (not counting potatoes, cassava, or yam)? (responses: 0 times a day, 1 time a day [at lunch or dinner], 2 times a day [at lunch and dinner], or 3 times or more a day); (c) In general, how many glasses a day do you have of natural fruit juice? (answers: 0 cup, 1 cup, 2 cups, or 3 cups or more); (d) In general, how many times a day do you eat fruits? (responses: 0 times a day, 1 time a day, 2 times a day, 3 times or more a day). The variable was calculated by summing the total number of daily portions of lettuce and tomato salad, or salad of any other vegetable or raw vegetable (a, maximum of 3), vegetables or cooked vegetables, not counting potatoes, cassava, or yams (b, at most 3), natural fruit juice (c, maximum of 3) and fruit (d, maximum of 3). Thus, individuals met the recommended consumption of fruits and vegetables when the sum was of five or more portions.

(iv) Self-reported AH (no or yes), obtained by the question: “Has any doctor ever diagnosed you as having high blood pressure?”;

(v) Self-reported hypercholesterolemia (yes or no), obtained through the question: “Has any doctor ever diagnosed you as having high cholesterol?”;

(vi) Nutritional status: Individuals were classified as normal (BMI 18.4 to 24.9 kg/m^2^), overweight (BMI 25 to 29 kg/m^2^), or obese (BMI > 30 kg/m^2^) [22]. To calculate BMI, weight and height measurements were made using commercial appliances such as portable electronic balances and portable stadiometers, respectively. All the equipment was calibrated, and the users were trained to standardize all measurements.

### 2.3. Statistical Analysis

Data were analyzed using the STATA software (Stata Statistical Software: Release 14, StataCorp LP., College Station, TX, USA) [23]. All analyses were performed using the routines for complex samples. A descriptive analysis of the sociodemographic variables was performed using absolute and relative frequency with 95% confidence intervals (95% CI). 

Bivariate and multiple analysis was performed to verify the magnitude of the association between the dependent and the independent variables. Bivariate Poisson regression was used to verify the association between the dependent variable of DM and each of the independent variables. Variables with a *p*-value < 0.20 were included in a Poisson multiple regression model [24,25]. The results are presented as crude prevalence ratio (PR), adjusted prevalence ratio (PRadj), and their respective 95% CIs. Statistical significance was established by the Wald Chi-square test. Values of *p* < 0.05 were considered statistically significant.

### 2.4. Ethical Aspects

This study was approved by the National Commission of Ethics in Research, opinion number 328,159/2013. Written consent was obtained from all participants.

## 3. Results

There were 1774 adults recruited in the State of Maranhão by the NHS. The mean age of participants was 41.09 years (95% CI: 39.61–42.57). About half of the individuals were female (52.14%). The majority of the participants lived with a spouse (61.70%), had low education identified as an incomplete primary education or no education (60.97%), and self-described their race/skin color as brown (67.0%).

The prevalence of self-reported DM in Maranhão was 5.39% (95% CI: 3.73%–7.73%). Table 1 shows the potential factors associated with DM from the sample investigated in the bivariate analysis. The prevalence was significantly higher in women than men (7.7% versus 2.9%, PR: 2.58, *p*-value = 0.021). In addition, prevalence increased with advancing age, ranging from 0.9%, in the age group 18 to 39 years old, to 17.2%, in individuals aged ≥ 60 years. The self-reported DM prevalence was 7.8 times higher (PR: 7.8; *p*-value = 0.002) in adults aged 40–59 years, and 18.6 times higher (PR: 18.6; *p*-value < 0.001) in older adults, when compared to the age group of 18 to 39 years.

The medical diagnosis of DM was more frequently reported in adults with lower levels of education (no education or only having complete primary education) (7.4% versus 1.7%, PR: 4.35, *p* = 0.039). No statistical difference was found between the DM prevalence and the race/skin color categories (*p* > 0.05).

Regarding risk factors, it was verified that the prevalence of DM was higher in individuals with self-reported AH (13.4% versus 4.1%, PR: 3.25, *p* < 0.001) and self-reported hypercholesterolemia (27.2% versus 2.9%, PR: 9.5, *p* < 0.001). The prevalence was higher among former smokers than in adults who never smoked (9.6% versus 4.3%, PR: 2.23, *p* = 0.017).

Table 2 shows the regression analysis of risk factors for DM in Maranhão. After adjusting the regression model, it was verified that age ≥ 60 years (PRadj: 4.79, *p*-value < 0.001), female gender (PRadj: 2.31, *p*-value = 0.026), low level of education (PRadj: 4.36, *p*-value = 0.039), and self-reported AH (PRadj: 2.77; *p*-value = 0.034) were independently associated with the outcome.

## 4. Discussion

The present study analyzed the prevalence and factors associated with DM in a sample of 1774 adults in the State of Maranhão, Northeastern Brazil, using data from the 2013 NHS. Self-reported DM prevalence in adults was estimated at 5.39%, with the frequency being higher in women than in men. Multiple regression analysis showed that age ≥60 years, female sex, low education, and presence of self-reported AH were factors associated with the analyzed outcome. Moreover, there was a high prevalence of risk factors for NCDs, such as obesity, alcohol, and tobacco use, although there was no statistical difference with the analyzed outcome.

The prevalence of self-reported diabetes estimated in this study (5.39%) was similar to that estimated for the Brazilian adult population, according to the NHS data for 2013 (6.2%) [6]. Population-based studies on the prevalence of diabetes in the Northeast Region of Brazil are scarce. However, the Vigitel study of 2013 in São Luís estimated a self-reported DM prevalence of 4.9% (95% CI: 3.8–6.0), being similar to that found in this investigation [26]. Another investigation conducted in the population of São Luís estimated a self-reported diabetes prevalence of 5.1% (95% CI: 3.8–6.9), which is also similar to this investigation [27]. 

The factors associated with DM in São Luís were similar to those found in other population-based studies [6,10,27,28,29]. The present study showed a self-reported DM prevalence is almost twice as high in women when compared to men. Other analyses of population-based studies using self-reported DM as an indicator found similar results [2,26].For example, Vigitel data in São Luís estimated that the DM prevalence was almost triple the amount for women than in men (6.4 versus 3.1) [26]. This fact may be related to the higher proportion of women seeking preventive health services, which may contribute to the increased frequency of medical diagnosis in this subgroup [6].

In this investigation, age was a demographic factor associated with DM, and is consistent with other studies [30]. In older adults, DM represents an alarming public health problem in developed and developing countries, leading to frequent acute and chronic complications, mortality, and increased risk of institutionalization [31,32]. This fact can be attributed to the cumulative risk of biological, social and behavioral risk factors throughout life [31,32]. Increasing age leads to a decrease in insulin secretion and functional decrease of the pancreatic islets. In addition, central obesity, physical inactivity, sarcopenia and changes in body composition may lead to increased insulin resistance, potentiating the risk of metabolic syndrome and DM [31,33]. 

There was an association between low levels of education and self-reported DM in the present study, even after adjusting for other potential confounders. In fact, one study observed that Brazilian adults with low education levels have a high prevalence of diabetes and other chronic diseases, such as hypertension, when compared to those with higher education levels (high school level or higher) [34]. A low education level may decrease diet quality and potentiate physical inactivity and unhealthy behaviors [35,36]. In addition, individuals with a low education level have less knowledge about healthy habits and practices, as well as access to prevention services, which increases their vulnerability to diabetes.

Self-reported AH was independently associated with DM, as well as in other investigations [6,10,27]. There is an important overlap between DM and hypertension in terms of etiology and causal mechanisms [37]. The presence of both diseases accelerates a decrease in renal function, and the development of retinopathy and cerebral diseases; presenting high burden for patients and the health services [38]. Common pathways for both diseases include central obesity, physical inactivity, inadequate nutrition, inflammation, oxidative stress, and insulin resistance [37]. In addition, patients with DM have increased peripheral arterial resistance caused by an increased volume of body fluids that raises systemic blood pressure and potentiates the risk of AH [39]. Thus, actions aimed at prevention and control of DM can also lead to a reduction in the burden of other NCDs, such as AH.

The present study has some limitations. First, the transverse nature of the investigation does not allow for establishing a cause and effect relationship between DM and the independent variables analyzed [40]; longitudinal studies are needed to verify this causality. Second, data related to DM were self-reported, which may have caused underestimating the magnitude of this condition. However, some studies have shown that self-reporting of DM corresponds to a good indicator of evaluation in the general population [6,28,41]. Data related to the use of elicit substances and eating habits have also been reported, which could lead to memory and response bias. 

## 5. Conclusions

In conclusion, the present study found the self-reported DM prevalence to be similar to the estimated prevalence of Brazil’s general population. Low education, female gender, and obesity were the associated factors of DM in Maranhão. The results of this investigation demonstrate the need to improve monitoring of this disease in Maranhão. Therefore, public health policy development is necessary to reduce diabetes burden in this region, with focused attention on groups who are vulnerable to developing this disease.

## Figures and Tables

**Table 1 ijerph-16-00047-t001:** Prevalence of diabetes mellitus (DM) according to sociodemographic variables and potential risk factors in the adult population of Maranhão, National Health Survey, 2013.

Variables	Total	Diabetes		Crude PR ^2^ (95% CI) ^1^	*p*-Value ^3, 4^
		*N*	% (95% CI) ^1^		
Sociodemographic					
Gender					
Male	750	24	2.7 (1.6–5.4)	1.00	
Female	1.024	61	7.7 (5.0–11.3)	2.58 (1.17–5.72)	0.021
Age (years)					
18–39	956	8	0.9 (0.4–2.4)	1.00	
40–59	566	35	7.3 (4.6–11.4)	7.80 (2.4–25.6)	0.002
≥60	252	43	17.5 (10.7–27.2)	18.6 (5.5–63.0)	<0.001
Marital status					
Lives with a partner	1.014	50	6.2 (4.4–8.6)	1.00	
Lives without a partner	760	35	4.0 (2.0–7.7)	0.65 (0.32–1.29)	0.208
Race/skin color					
White	355	19	4.0 (2.1–7.4)	1.00	
Brown	1.098	52	5.9 (3.9–8.7)	1.47 (0.65–3.34)	0.333
Black	304	14	5.0 (2.0–11.8)	1.26 (0.37–4.30)	0.696
Education					
Complete/incomplete higher education	141	7	1.7 (0.5–5.6)	1.00	
Secondary education	546	13	2.1 (1.1–4.1)	1.23 (0.35–4.26)	0.727
Complete primary education	264	12	3.0 (1.1–7.8)	1.75 (0.62–4.91)	0.272
Incomplete primary education/no education	823	53	7.4 (4.6–11.5)	4.35 (1.08–17.42)	0.039
Area					
Capital	854	42	4.1 (3.6–4.7)	1.00	
Metropolitan region	181	9	5.1 (3.8–6.8)	1.24 (0.89–1.69)	0.186
Others	739	34	5.6 (3.8–8.3)	1.36 (0.90–2.06)	0.129
(Type of) Residence					
Urban	1.228	63	6.1 (4.0–8.8)	1.00	
Rural	546	22	3.9 (2.2–6.6)	0.64 (0.32–1.21)	0.160
Substance use ^5^					
Regular consumption of alcohol					
No	1.417	74	6.2 (4.0–9.2)	1.00	
Yes	357	11	2.7 (0.8–9.2)	0.44 (0.09–1.94)	0.267
Heavy alcohol consumption					
No	1.557	78	5.6 (3.9–8.2)	1.00	
Yes	217	7	3.4 (0.6–16.8)	0.60 (0.09–3.80)	0.576
Tobacco use					
Never	1.294	48	4.3 (2.5–7.2)	1.00	
Former smoker	277	30	9.6 (6.5–14.0)	2.23 (1.17–4.24)	0.017
Smoker	203	7	5.3 (2.1–13.0)	1.23 (0.43–3.50)	0.676
Eating habits					
Recommended consumption of fruits and/or vegetables					
No	1.376	57	5.2 (3.4–7.8)	1.00	
Yes	398	28	6.2 (3.9–9.7)	1.19 (0.60–2.36)	0.595
Comorbidities					
Hypertension					
No	1.528	49	4.1 (2.8–6.0)	1.00	
Yes	246	36	13.4 (8.2–20.9)	3.25 (1.94–5.46)	<0.001
Hypercholesterolemia					
No	1.597	48	2.9 (2.0–4.1)	1.00	
Yes	177	37	27.2 (17.3–40.1)	9.46 (5.97–15.00)	<0.001
Nutritional status					
Eutrophic	793	33	5.3 (3.3–8.3)	1.00	
Overweight	592	31	6.6 (3.6–11.8)	1.25 (0.66–2.37)	0.484
Obesity	309	21	4.5 (2.5–8.0)	0.85 (0.38–1.89)	0.689

^1^ 95% confidence interval; ^2^ Prevalence ratio; ^3^ Wald Chi-square test; ^4^ Refers to the association of each variable with the outcome (DM); ^5^ Last 30 days.

**Table 2 ijerph-16-00047-t002:** Multiple regression analysis of risk factors for diabetes mellitus (DM) in the adult population of Maranhão, National Health Survey, 2013.

Variables	PRadj ^1^	95% CI ^2^	*p*-Value ^3^
Age ≥ 60 years	4.79	2.04–11.23	<0.001
Female gender	2.31	1.10–4.86	0.026
Education: Incomplete primary education/no education	4.36	1.09–17.49	0.039
Arterial hypertension	2.77	1.08–7.10	0.034

^1^ Adjusted prevalence ratio; ^2^ 95% confidence interval; ^3^ Wald’s statistics; Note: Poisson model adjusted for age, gender, smoking, education, hypertension, and hypercholesterolemia.

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
