# Peer review of "Epidemiology of Self-Reported Diabetes Mellitus in the State of Maranhão, Northeastern Brazil: Results of the National Health Survey, 2013"

_ijerph, 2018, doi:10.3390/ijerph16010047_

Round 1
Reviewer 1 Report
- Line 35- 1.5 million deaths per year?
- Line 59: the abbreviation NCD and subsequent use of CNCD do not match
-Line 82-83: what is the justification for just looking at Maranhao versus the entire data set on Brazil?
-Line 100- are the listed geographical areas all applicable to Maranhao
-line 105 - how was this eating habit assessed? yes or no to this amount of fruits and vegetables together? separate? how does this change into salad consumption that is later reported? please clarify
-111-112: was BMI self-report or assessed in-person?
-the p-value comparisons in table 1 need to be clarified. Is the the p-value for the association with diabetes for each variable or for comparisons such as male versus female?
Author Response
Goiânia, December 2018
Dear reviewer. Thank you for the careful evaluation of our manuscript.
- Line 35- 1.5 million deaths per year?
Answer: Sorry for the error. It refers to 1.5 million deaths in 2012.
- Line 59: the abbreviation NCD and subsequent use of CNCD do not match.
Answer: Sorry for the error. The term was standardized throughout the text for NCDs.
-Line 82-83: what is the justification for just looking at Maranhao versus the entire data set on Brazil?
Answer: Thanks for the review. The data from Brazil, were published consolidated in a previous publication (Malta, DC, Bern, RTI, Iser, BPM, Szwarcwald, CL; Duncan, BB; Schmidt, MI Factors associated with self-reported diabetes according to the National Health Survey, 2013. Rev Public Health 2017, 51, 12s). Once we present the disaggregated data for Maranhão, we are trying to study the specific risk factors in this State, which corresponds to one of the poorest in the country.
-Line 100- are the listed geographical areas all applicable to Maranhao
Answer: yes.
-line 105 - how was this eating habit assessed? yes or no to this amount of fruits and vegetables together? separate? how does this change into salad consumption that is later reported? please clarify
Answer: Sorry for the mistake. It was presented the form of measurement of this variable in the session methods, with source duly referenced.
-111-112: was BMI self-report or assessed in-person?
Answer: The BMI was calculated. To calculate BMI, weight and height measurements were made using commercial appliances. We used, respectively, such as portable electronic balances and portable stadiometers, respectively. All the equipment was calibrated, and the trained ones were trained to standardize the measurement of theall measurements.
-the p-value comparisons in table 1 need to be clarified. Is the the p-value for the association with diabetes for each variable or for comparisons such as male versus female?
Answer: Sorry. The p values refer to the association of each variable with diabetes (bivariate analysis). We clarify this in the footnote to Table 1.
The authors

Reviewer 2 Report
There are several main concerns,
1) The diagnosis of diabetes mellitus is based on self-report, with the new case of diabetes missing, therefore the title should be revised to be “Epidemiology of self-reported diabetes mellitus…”.
2) Abstract. After adjustment in the regression model, DM was statistically associated with aging > 60 years, females, low educational level and comorbidity with hypertension”. Please address which confounders are adjusted.
3) Results. Several potential confounding factors were listed in Table 1; However most of these are not included as confounders in Table 2. Please explain.
4) For the multiple variable analysis, when category variable “Age > 60 years is independent variable, did the age still be confounders?
Author Response
Goiânia, December 2018
Dear reviewer.
Thank you for the careful evaluation of our manuscript.
We do all the editing of the English language.
There are several main concerns,
1) The diagnosis of diabetes mellitus is based on self-report, with the new case of diabetes missing, therefore the title should be revised to be “Epidemiology of self-reported diabetes mellitus…”.
Answer: Thanks for the suggestion. We did the rectification of the title.
2) Abstract. After adjustment in the regression model, DM was statistically associated with aging > 60 years, females, low educational level and comorbidity with hypertension”. Please address which confounders are adjusted.
Answer: Thanks for the suggestion. We did the change of summary and text indicating the confounding factors and adjusted variables: age, sex, smoking, education, hypertension and hypercholesterolemia.
3) Results. Several potential confounding factors were listed in Table 1; However most of these are not included as confounders in Table 2. Please explain.
Answer: thanks for the note. We include, as well as in previous studies (Malta, DC, Bern, RTI, Iser, BPM, Szwarcwald, CL; Duncan, BB; Schmidt, MI Factors associated with self-reported diabetes according to the National Health Survey, 2013. Rev Public Health 2017, 51, 12s). only variables with p-value <0.20 in the bivariate analysis and sex and age, independent of the b-value of bivariate by its great condoning factor. Also, we followed the recommendations of authors who say to fit in the regression model only variables with p-value <0.20.
4) For the multiple variable analysis, when category variable “Age > 60 years is independent variable, did the age still be confounders?
Answer: Answer: yes. Age was considered a confounding variable with hypertension and hypercolestolemia. At the same time, it is a predictor, influencing the outcome of the study. In summary, our Poisson statistical modeling was performed with the following variables: age, gender, smoking, education, hypertension and hypercholesterolemia. The command used in the stata was svy: poisson diabetes i.age i.gender i.smoking i.education i.hypertension i.hyperchol, irr.
The authors

Round 2
Reviewer 1 Report
No specific comments
Author Response
thank you for your comments